# Overexpression of *DoBAM1* from Yam (*Dioscorea opposita* Thunb.) Enhances Cold Tolerance in Transgenic Tobacco

**DOI:** 10.3390/genes13122296

**Published:** 2022-12-06

**Authors:** Lingmin Zhao, Yanfang Zhang, Ying Shao, Linan Xing, Mingran Ge, Xiuwen Huo

**Affiliations:** 1Horticulture Department, Faculty of Horticulture and Plant Protection Science, Inner Mongolia Agricultural University, Hohhot 010019, China; 2Inner Mongolia Academy of Agricultural and Animal Husbandry Sciences, Hohhot 010031, China

**Keywords:** yam, β-amylase, *DoBAM1*, starch degradation, cold tolerance

## Abstract

β-amylase (BAM) plays an important role in plant development and response to abiotic stresses. In this study, 5 *DoBAM* members were identified in yam (*Dioscorea opposita* Thunb.). A novel β-amylase gene *BAM1*, (named *DoBAM1*), was isolated from yam varieties Bikeqi and Dahechangyu. The open reading frame (ORF) of *DoBAM1* is 2806 bp and encodes 543 amino acids. Subcellular localization analysis indicates that DoBAM1 localizes to the cell membrane and cytoplasm. In the yam variety Dahechangyu, the starch content, β-amylase activity, and expression of *DoBAM1* were characterized and found to all be higher than in Bikeqi. *DoBAM1* overexpression in tobacco is shown to promote the accumulation of soluble sugar and chlorophyll content and to increase the activities of peroxidase (POD), superoxide dismutase (SOD), catalase (CAT), and β-amylase. Under cold treatment, we observed the induced upregulation of *DoBAM1* and lower starch content and malondialdehyde (MDA) accumulation than in WT plants. In conclusion, these results demonstrate that *DoBAM1* overexpression plays an advanced role in cold tolerance, at least in part by raising the levels of soluble sugars that are capable of acting as osmolytes or antioxidants.

## 1. Introduction

Yam (*Dioscorea opposita* Thunb.) belongs to the genus *Dioscorea* and the family Dioscoreaceae. In addition to its significance as a traditional starchy staple food, many *Dioscorea* species have been fostered and used extensively for nutritional and medicinal aims [1]. Yam is cultivated in many temperate and tropical regions and is one of the ten most significant rhizome crops in the world [2]. The tubers of this crop have high edible value, are highly nutritious, and provide additional health benefits. The secondary metabolites abundant within yam tubers can reduce blood lipid levels and inhibit cholesterol, fat absorption, cancer cell growth, and neurodegeneration [3,4]. Starch, as the most abundant component of yam tubers, contributes greatly to the value of all yam-tuber-related foodstuffs [5,6].

Starch is the main energy-storage carbohydrate and is closely associated with abiotic stresses and secondary metabolism pathways in plants [7,8,9]. β-amylase (BAM) is a critical enzyme regulating starch metabolism, with distinctive roles in the breakdown of starch and the synthesis of sugar [8]. BAMs (β-amylases; EC 3.2.1.2) are members of the glycosylhydrolase family 14 (PF01373). The important role of these enzymes is conferred by two typical catalytic sites at the *N*-terminus that are central to hydrolyzing the 1,4-α-glucosidic linkages within starch [10,11]. Hence, BAMs are thought to positively regulate the accumulation of soluble sugars. In addition, soluble sugars are produced during the degradation of starch, and the synthesis of soluble sugars is transferred from chloroplasts to the cytoplasm, which can then participate in the energy metabolism pathway [12,13]. In potato, the activity of BAM increases four- to five-fold at low temperatures, causing sweetness in tubers [14]. In banana, *MaBAMs* participate in the regulation of fruit-ripening, and the expression levels of several *MaBAMs* change markedly under several stress conditions, including low-temperature stress conditions [15]. Studies in perennial plants, such as pear, blueberry, orange, and tea, have shown that the expression level of *BAM* and activity of BAMs can both be increased under cold stress conditions [16,17,18].

In plants, cold stress can lead to a range of metabolic changes. In plants, cold conditions affect particular physiological, biochemical, and molecular processes, enable tolerance to cold stress, and determine geographic distribution [19,20]. Under cold stress, reactive oxygen species (ROS) accumulate as the balance between ROS production and ROS scavenging is disrupted, which may lead to cell membrane damage and osmotic imbalance [21,22]. Cold stress also counteracts the defense pathway and can lead to increased accumulation of osmolytes or osmoprotectants, such as proline and soluble sugars, via the upregulation of genes related to metabolism [23,24,25]. There is mounting evidence showing that soluble sugars are of great importance in ROS scavenging and in enhancing cold tolerance [26,27]. For example, carbohydrates in potato leaves undergo significant changes during the early phases of cold stress, leading to the accumulation of soluble sugars [28]. In wheat, defense against low-temperature stress is achieved through antioxidant enzyme activity [29].

Chinese yam is unique because it is native to temperate regions and can tolerate much colder temperatures than its relatives [30]. Both Dahechangyu and Bikeqi are early-maturing varieties that grow in the Inner Mongolia Autonomous Region, north of the Yangtze River Basin in China, and are widely cultivated. Continuous crop damage, pests, and diseases in Southern China severely restrict the development of traditional yam production areas in the country. Therefore, in 2009, China’s Ministry of Finance and Ministry of Agriculture officially proposed the implementation of a “south to north migration of yam” and “north to west expansion” strategy (200903022) [31]. Thus far, areas in Northwest China, such as Yulin City of Shaanxi Province [32] and Xinjiang Autonomous Region [33], have been widely planted. The western region of Inner Mongolia has become an emerging yam-producing area; it is a cold and arid region with abundant land resources, sufficient light, an arid climate, low air humidity, a large temperature difference between day and night, a loose soil structure, and fewer pests and diseases, which are ideal for yam cultivation: the yams produced here are also of higher quality. Nevertheless, the annual frost-free period is only 100–150 days, and the average temperature throughout the year is 2–14 °C. The growth cycle of yam lasts 180–200 days from early April to mid-October [1]. Therefore, the low temperature is a key factor limiting the development of the yam industry. Under such conditions, we studied *DoBAM1*, a key gene regulating cold stress, as a candidate for molecular breeding aimed at extending the yam growth cycle through the breeding of mid- to late-maturing varieties for cultivation in northern regions.

In this work, we isolated a novel *BAM* gene called *DoBAM1* from the yam varieties Bikeqi and Dahechangyu. We overexpressed *DoBAM1* in tobacco and assessed the phenotype and physiological changes to determine the characteristics that improve cold tolerance in addition to characterizing the expression of stress-response genes. We found a correlation between increased BAM activity and the higher accumulation of soluble sugars after cold treatment. This study provides a theoretical basis for further investigations on the role of *BAM* in development and stress tolerance in yam.

## 2. Materials and Methods

### 2.1. Plant Materials, Cultivation, and Stress Treatments

Two yam cultivars with different starch content were used as the experimental materials. Dahechangyu (DHCY) is a high-starch yam cultivar, while Bikeqi (B1) is a low-starch yam cultivar. The two cultivars were planted in a completely randomized block design at the Germplasm Resource Nursery of Inner Mongolia Agricultural University (40°29′28.01″ N and 111°47′07.69″ E, Hohhot, China). Yams of each cultivar were randomly sampled in the morning at five different developmental stages: after 105 d (T1), 120 d (T2), 135 d (T3), 150 d (T4), and 165 d (T5). Micropropagated plantlets of DHCY were selected on Murashige and Skoog 1962 (MS) culture media containing 7.0 g/L agar and 30 g/L sucrose (pH 6.0). Plants grown in bottles for thirty days were selected based on uniformity of growth and treated in a low-temperature incubator at 4 °C. Whole leaves were then sampled at 0, 1, 6, 12, and 24 h time points. The tubers, stems, and young leaves of yams were stored at −80 °C for further use. Three biological replicates of samples were used for the purposes of RNA extraction and physiological index assay.

### 2.2. Determination of Physiological Parameters

DHCY and B1 yam cultivars grown in the field for different sampling periods were harvested, heated at 80 °C for approximately 24 h, and then ground into powder to obtain a constant dry weight (DW). The dried samples (100 mg) were used to determine the starch content, as described by Ning Li et al. [34]. The fresh samples (1 g) were used to measure the β-amylase enzyme activity using a β-amylase assay kit (Nanjing Jiancheng Bioengineering Institute, Nanjing, China).

### 2.3. DoBAM1 Expression Analysis

Total RNA was extracted from tubers, stems, and young leaves in the field and leaves under 4 °C treatment for the synthesis of the first-strand cDNA. DoBAM1-specific primers, DoBAM1-qF1/R1, were designed to determine expression levels (Appendix A). The PCR thermal cycle conditions were as follows: initial denaturation at 95 °C for 30 s; 45 cycles of denaturation at 95 °C for 5 s; annealing at 60 °C for 30 s. Melting-curve analysis involved 1 cycle of 95 °C for 5 s, 60 °C for 1 min, and 95 °C for 15 s. In addition, all data were reckoned using the 2^−ΔΔCt^ method [35]. The actin gene was used as an internal control (Appendix A) [36].

### 2.4. Isolation and Cloning of the DoBAM1 Gene

The gene BAM1 (Unigene0000684) was screened according to the transcriptome results obtained for yam via our previous high-throughput sequencing study [37,38]. Total RNA was isolated from tuber using a MiniBEST Plant RNA Extraction Kit (Takara, Beijing, China). The rapid amplification of cDNA ends (RACE) method (SMARTer^®^ 5′RACE & 3′RACE Kit, Takara, Kusatsu, Japan) was used to clone the full-length sequence of *BAM1* cDNA. Sequencing results were compared and spliced with the conserved region sequence via DNAMAN 6.0. The 5′/3′RACE clone primer sequences DoBAM1-5′R1/3′F1 and the open reading frame (ORF) primers DoBAM1-ORF-F/R were designed using Primer Premier 5.0 (Appendix A). The PCR thermal cycle conditions were as follows: initial denaturation at 94 °C for 10 min, 30 cycles of denaturation at 94 °C for 30 s, annealing at 56 °C for 30 s, and extension at 72 °C for 2 min, with a final extension at 72 °C for 10 min. Subsequently, we inserted the pGM-T vector (TIANgel, Beijing, China) for sequencing.

### 2.5. Sequence Analysis of DoBAM1

The full-length cDNA of DoBAM1 was analyzed using the online BLAST tool of the National Center for Biotechnology Information (NCBI, http://www.ncbi.nlm.nih.gov/, accessed on 25 August 2022). The accession numbers of the Arabidopsis thaliana (*A. thaliana*) *BAM* genes were referred from the research of Fulton et al. [12,39], and the full-length coding sequences (CDS) of genes were downloaded from TAIR (https://www.arabidopsis.org/, accessed on 28 August 2022). BAM protein sequences of rice (*Oryza sativa*), water yam (*Dioscorea alata*), and potato (*Solanum tuberosum*) were downloaded from the Phytozome v13 database. DNAMAN software (Lynnon Biosoft, San Ramon, CA, USA) was used for the multiple alignment analysis of *A. thaliana* and yam sequences. Based on this, the phylogenetic analysis of the BAM gene family members from yam and other species was conducted with the MEGA6 software package using the neighbor-joining method and 1000 bootstrap replications [40]. The ProtParam tool (http://web.expasy.org/protparam/, accessed on 26 August 2022) was used to calculate the theoretical molecular weight (kDa) and isoelectric point (pI). (http://web.expasy.org/protparam/, accessed on 26 August 2022).

### 2.6. Subcellular Localization of DoBAM1

The ORF of DoBAM1 was amplified using the specific primers DoBAM-SF/R with restriction enzyme cutting sites and was then inserted into the CaMV-35S-GFP vector (Appendix A). Following sequence confirmation, the 35S::DoBAM1-GFP and CaMV-35S-GFP (control) plasmids were transferred into *Agrobacterium tumefaciens* EHA 105 (Weidi Biotechnology, Shanghai, China). The empty CaMV-35S-GFP vector was used as a control. The vectors were transferred into tobacco (*Nicotiana tabacum* L.) leaves [17], which were then cultured for 2–3 days. Infected parts of leaves were then cut and used for fluorescent signal detection under a laser confocal microscope (C2-ER; Nikon, Tokyo, Japan).

### 2.7. Generation and Identification of Transgenic Plants

To construct the pPZP221–DoBAM1 expression vector, the DoBAM1 CDS with the *BamH*Ⅰ and *Kpn*Ⅰ restriction sites was amplified using the specific primers DoBAM-ZF/R (Appendix A) and was inserted into the pPZP221-35S-NOS expression vector. The Agrobacterium tumefaciens strain EHA101 (Weidi Biotechnology, Shanghai, China) was transformed into wild-type (WT) tobacco via the freeze–thaw method. Transgenic plants were grown on 1/2 MS containing 50 mg/L gentamicin and 250 mg/L cefotaxim. DNA and RNA of transgenic and WT tobacco leaves were isolated using a Hi-DNAsecure Plant Kit (TIANGEN, Beijing, China). Finally, we selected three positive transgenic lines (BAM4, BAM5, and BAM12) for further gene-functional verification.

### 2.8. Cold Treatment of Transgenic Tobacco and Analysis of Physiological Characteristics

For the cold tolerance assay, 3-week-old WT and transgenic tobacco in 1/2 MS were transferred to individual pots containing substrate soil for a period of 2 weeks. Following the method described by Guo [41] as a reference, we carried out treatments at 4 °C (T1) and −2 °C (T3) in addition to 2 °C (T2) for 2 h to determine the cold tolerance of transgenic tobacco. To measure physiological changes, the healthy, uniform-sized plants from the WT and transgenic lines were selected for the establishment of lines for further analysis. Starch content was then determined using the method of Wang et al. [42]. Soluble sugar and chlorophyll content was assessed using the method described by Su [43] and Chow [44]. SOD, POD, CAT, and BAM activity, and MDA content, were all measured according to the kit manual (Nanjing Jiancheng Bioengineering Institute, Nanjing, China). Histochemical staining with 3,3′-diaminobenzidine (DAB) and nitroblue tetrazolium chloride (NBT) was used to examine the in situ accumulation of hydrogen peroxide (H_2_O_2_) and superoxide (O_2_^¯^) under cold stress [45,46].

### 2.9. Statistical Analysis

All data were based on three independent samples. Data and graphs were processed using Excel 2019. The statistical analysis was performed using the SPSS 21.0 software package. Differences among treatments were identified by data analysis using t-tests and one-way ANOVA, and Duncan’s multiple comparisons were used for sample comparisons at significance levels of *p* < 0.05 or *p* < 0.01.

## 3. Results

### 3.1. Analysis of Starch Content and Amylase Activity in Tubers of Yam Cultivars

Tubers from the two yam cultivars were harvested at five ripening stages to investigate the changes in starch content and β-amylase activity. As the tubers developed, the starch content of both DHCY and B1 cultivars increased and peaked at T3. However, at each stage of development, the starch content in DHCY was significantly higher than in B1 (Figure 1A). We also found a significant difference in β-amylase activity between the two yam cultivars in the T1–T3 period, although there was a decrease in both cultivars after the T2 stage (Figure 1B).

### 3.2. Expression Patterns of DoBAM1

The transcripts of *DoBAM1* reached their highest levels at the T2 stage. The expression of *DoBAM1* was lowest at T5 (Figure 2A). The results of tissue-specific expression analysis show that the levels of *DoBAM1* were highest in leaves (Figure 2B). The expression levels were lowest in B1 when considering tubers and lowest in DHCY when considering stems. The transcript level of *DoBAM1* first increased and then decreased during the twenty-four hours of cold treatment, but it remained higher throughout than the value at the initial zero-hour time point (Figure 2C). To conclude, the results indicate that *DoBAM1* is responsive to cold stress.

### 3.3. Isolation and Characteristics of DoBAM1

In a previous study, we obtained and analyzed the *BAM* gene (GenBank Accession No. MT683336) from the transcriptome database (unpublished data) of *Dioscorea opposita* Thunb. The complete ORF of 2806 bp is flanked by a 1091 bp 5′-untranslated region (UTR) and a 42 bp 3′-UTR (Figure 3). The cDNA, designated as *DoBAM1*, contains an intact ORF and encodes a polypeptide of 543 amino acids with a calculated molecular mass of 622.92 kD and an isoelectric point of 5.60. As predicted by SWISS-MODEL, the secondary structure of the DoBAM1 protein contains 37.57% helices, 12.52% extended strands, and 6.54% turns. To analyze the evolutionary relationship of DoBAM1, a phylogenetic tree was constructed using a total of 32 BAM protein sequences from yam, *Arabidopsis*, water yam, potato, and rice (Figure 4). The BAM proteins were divided into four clades. DoBAM1 exhibited the highest homology with DaBAM5. The multisequence alignment revealed that the Glyco_hydro_14 domain and the catalytic residues Glu-186 and Glu-380 were highly conserved across yam and *Arabidopsis* BAM amino acids (Figure 5).

### 3.4. Subcellular Localization of DoBAM1

To further understand the function of DoBAM1 in the regulation of transcription, we constructed a DoBAM1–GFP fusion vector, as well as a GFP fusion expression empty vector and a control. These were then transformed into tobacco leaves, and GFP fluorescence was visualized under confocal microscopy. We observed that the DoBAM1–GFP was located in the cell membrane and cytoplasm, while fluorescence corresponding to expression from the control vector was distributed throughout the whole cell (Figure 6).

### 3.5. Regeneration and Identification of Transgenic Tobacco with DoBAM1 Gene

Overexpression of *DoBAM1* from the recombinant vector pPZP221–DoBAM was used to research the cold stress tolerance of *DoBAM1* in transgenic tobacco (Figure 7A). The transgenic tobacco exhibited increased growth under control conditions as well as higher tolerance of cold treatment than the WT plant (Figure 7B). The relative expression of *DoBAM1* in the three overexpressing lines was significantly higher than in the untreated transgenic plants (Figure 7C).

### 3.6. Physiological Characteristics of Tobacco Overexpressing DoBAM1 under Cold Stress

*DoBAM1* is highly affected by cold stress, so we measured the physiological characteristics of transgenic and WT plants to understand the mechanisms of its ability to enhance cold tolerance. Our results show that β-amylase activity was significantly higher in the three transgenic tobacco lines than in WT under the different cold treatments (Figure 8A). However, the starch content was significantly lower than in WT (Figure 8B). We found that the overexpression of *DoBAM1* induced the accumulation of soluble sugars under cold treatment and that the soluble sugar content was significantly higher than in the untreated transgenic plants (Figure 8C). Similarly, the chlorophyll content was significantly higher in the three overexpressing lines than that in the untreated transgenic tobaccos (Figure 8D).

Abiotic stress induces a high level of ROS production, which leads to cellular damage [47]. ROS scavenging is therefore essential to protect plants [48,49]. Under the different cold treatments, we examined the accumulation of H_2_O_2_ and O_2_^−^ in tobacco leaves by means of in situ histochemical staining using DAB and NBT. The stained leaves exhibited conspicuous differences under the different cold stressors. Serried plaques appeared less frequently in the leaves of transgenic tobacco, as evidenced by the DAB and NBT staining (Figure 9A–C). These results indicate that overexpressing *DoBAM1* decreases ROS accumulation under cold stress conditions. In plants, SOD, CAT, and POD activity and MDA content are all involved in the process of O_2_^−^ being reduced to H_2_O_2_, which enhances plant cold resistance [50,51]. We noticed that the *DoBAM1*-overexpressing lines displayed significantly higher SOD, POD, and CAT activities than WT (Figure 10A–C). However, the MDA content was significantly lower than in the WT plants (Figure 10D). These results indicate that the overexpression of *DoBAM1* strengthens the antioxidant capacity toward maintaining sustain the dynamic balance of ROS under cold stress conditions.

## 4. Discussion

Plants have developed the capacity to tolerate freezing temperatures to survive under conditions of cold stress [11]. It has been confirmed in numerous studies that starch is important in regulating their tolerance to stresses such as high salinity and low temperatures by releasing energy from sugar, obtaining metabolites produced from sugar, and through starch breakdown [9]. Starch breakdown can be used to provide energy and compensate for insufficient photosynthesis, and the metabolites formed by the synthesis and breakdown of sugars act as osmoprotectants under stress conditions [52]. A large spectrum of enzymes is implicated in the degradation of starch, including *BAMs* [17]. Recently, *BAMs* have been isolated and cloned in multiple plant species, such as *A. thaliana* [53], *Musa acuminata* (banana) [15], *Chenopodium quinoa* Willd. (quinoa) [54], *Ziziphus jujuba* Mill. (jujube) [55], *Oryza sativa.* (rice) [56], and others. In this study, we cloned the full-length sequence of *BAM1* from yam and named it *DoBAM1*. According to the phylogenetic analysis, a total of 5 *BAMs* are present in yam, and these are divided into four groups, suggesting that BAMs are evolutionarily conserved among different plants. DoBAM1 can be classified into the Group I category together with DaBAM5 from *Dioscorea alata* L. and OsBAM6 from *Oryza sativa.* Multiple sequence alignment has shown that DoBAMs and AtBAMs share similar amino acids related to substrate binding and hydrolytic catalysis [57]. In this study, subcellular localization indicated that DoBAM1 is mainly distributed in the cytoplasm and cell membrane, a finding not wholly consistent with previous findings [58]. In yam tubers, we found that the expression of *DoBAM1* and β-amylase was higher in the high-starch yam cultivar DHCY than in the low-starch cultivar B1. Based on yam growth in our region, DHCY in our experiment was treated with cold stress at 4 °C, and it was shown that the transcript expression levels of *DoBAM1* were significantly higher than those of the control. In light of the above, we speculate that *DoBAM1* also plays a crucial role in cold stress resistance. This supposition is confirmed by means of functional identification using *DoBAM1* transgenic tobacco.

Sugar metabolism and carbon partitioning are used to maintain the balance required for plant growth and also represent two effective methods to respond to abiotic stress conditions [59]. Soluble sugars, acting as antioxidants, have been shown to ameliorate ROS-derived oxidative stresses [27,60]. In our study, the three selected *DoBAM1*-overexpressing lines (BAM4, BAM5, and BAM12) exhibited a more robust phenotypic morphology, with higher BAM activity, higher soluble sugar levels, and significantly lower starch content than wild tobacco under cold-stress conditions. This indicates that the materiality of sugars derived from *DoBAM1* is involved in starch hydrolysis in the cold-tolerance process. Plants have evolved efficient scavenging mechanisms, including both enzymatic and non-enzymatic systems [61]. As one of the most important enzymes in abiotic stress, SOD can catalyze O_2_^¯^ conversion to O_2_ and H_2_O_2_, which are further scavenged by the coordinated action of POD with other factors [62,63,64]. In this study, the DAB and NBT staining results show that the accumulation of H_2_O_2_ and O_2_^−^ is greatly decreased in *DoBAM1*-overexpressing lines, compared with WT plants. Under different cold treatments, the activities of SOD, CAT, and POD were significantly higher in the *DoBAM1*-overexpressing plants than in WT. Numerous previous studies have confirmed that lower MDA content is indicative of enhanced abiotic stress tolerance [65,66,67]. In the present study, the MDA content was lower in DoBAM1-overexpressing lines than in WT. To conclude, our results demonstrate that *DoBAM1* overexpression enhances soluble sugar accumulation and regulates ROS homeostasis in response to cold stress, in line with the findings of previous studies [62,68]. The results of this study indicate that *DoBAM1* may be a reference by which to investigate the molecular mechanisms involved in cold tolerance, and it represents a potential gene resource for improving the abiotic resistance of yam and other crops.

## Figures and Tables

**Figure 1 genes-13-02296-f001:**
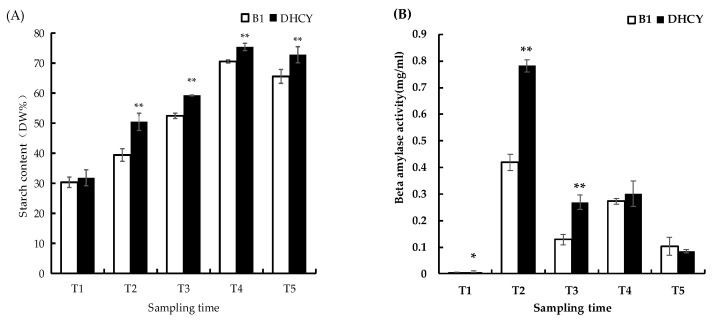
Changes in (**A**) starch content and (**B**) β-amylase activity in yam tubers. * and ** indicate a significant difference between the DHCY and B1 at *p* <0.05 and *p* < 0.01, respectively.

**Figure 2 genes-13-02296-f002:**
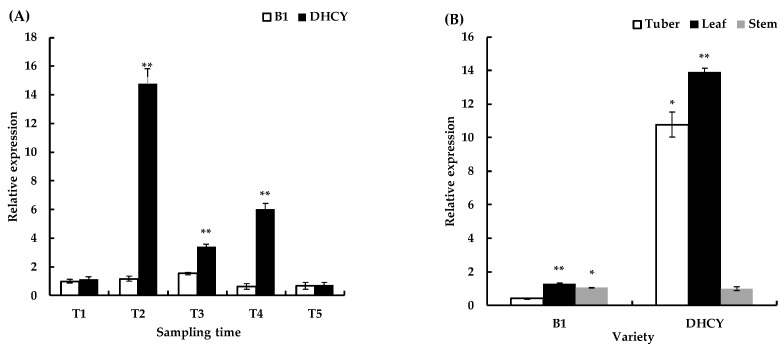
(**A**) Changes in *DoBAM1* expression during tuber expansion. (**B**) Expression of *DoBAM1* in different tissues of yam. (**C**) Expression levels in yam leaves during time course of cold stress. * and ** indicate a significant difference at *p* < 0.05 and *p* < 0.01, respectively.

**Figure 3 genes-13-02296-f003:**
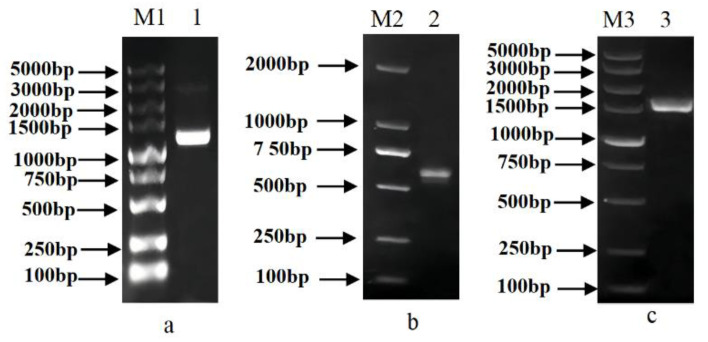
PCR amplification of *DoBAM1* of full-length cDNA. (**a**) M1: DL2 000 bp DNA Marker; 1: 5′RACE PCR; (**b**) M2: 2000 bp DNA Marker; 2: 3′RACE PCR; (**c**) M3: DL5 000 bp DNA Marker; 3: ORF full-length PCR.

**Figure 4 genes-13-02296-f004:**
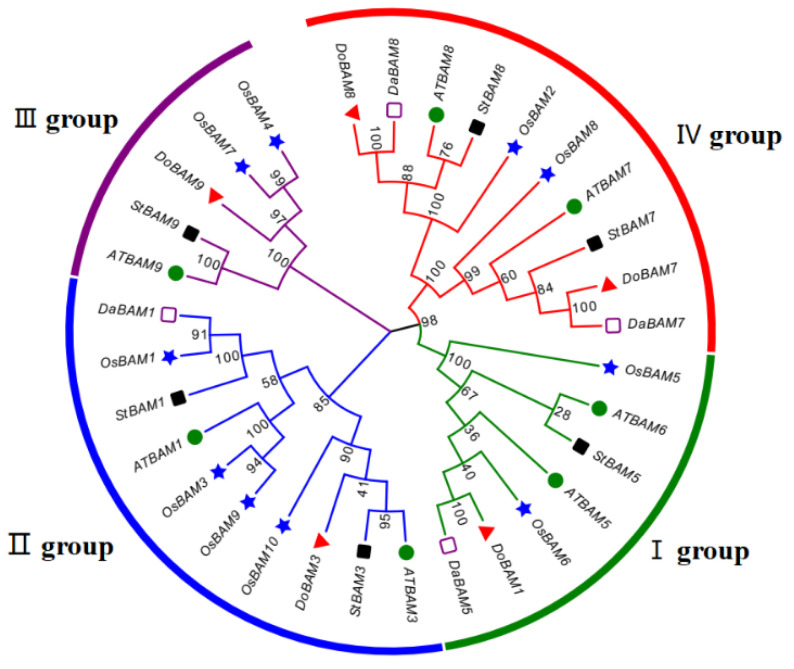
Phylogenetic analysis of DoBAM1 using 5 DoBAMs from yam, 7 AtBAMs from *Arabidopsis*, 4 DaBAMs from water yam, 6 StBAMs from potato, and 10 OsBAMs from rice. Red indicates IV group, green indicates I group, blue indicates II group, and purple indicates III group. The red data indicate DoBAM1.

**Figure 5 genes-13-02296-f005:**
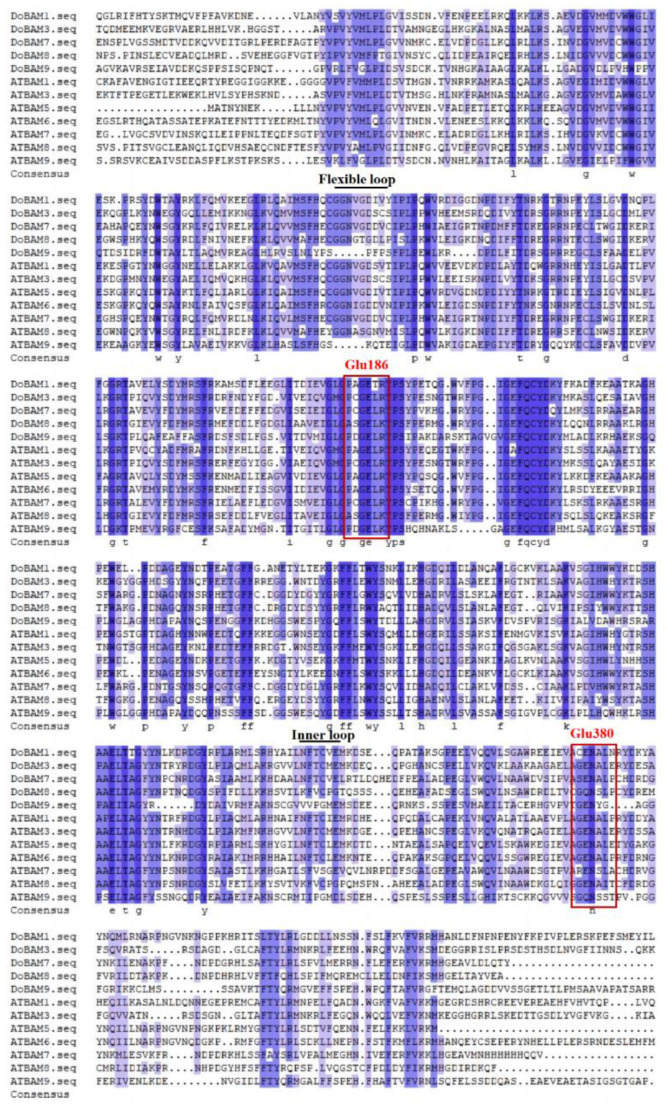
Multiple sequence alignment of DoBAM amino acid family. Blue shading indicates highly conservative substitutions. The two red boxes highlight the catalytic residues Glu-186 and Glu-380.

**Figure 6 genes-13-02296-f006:**
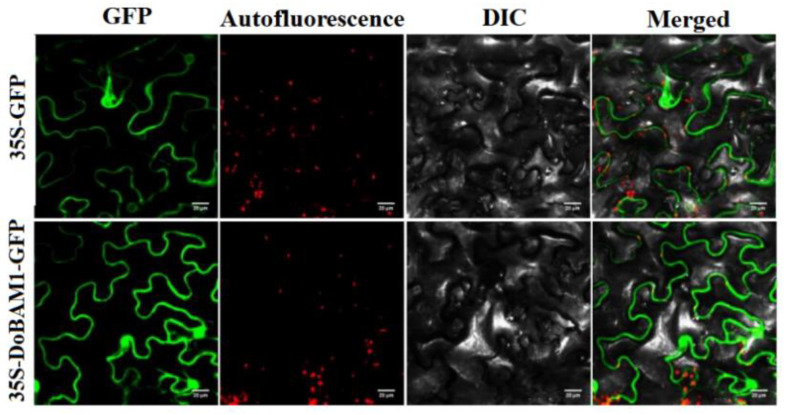
Subcellular localization of the DoBAM1 fusion protein in cells of tobacco leaves. The 4 panels from left to right indicate the GFP fluorescence, chloroplast autofluorescence, bright field, and merged images, respectively. The horizontal scale bar = 20 μM.

**Figure 7 genes-13-02296-f007:**
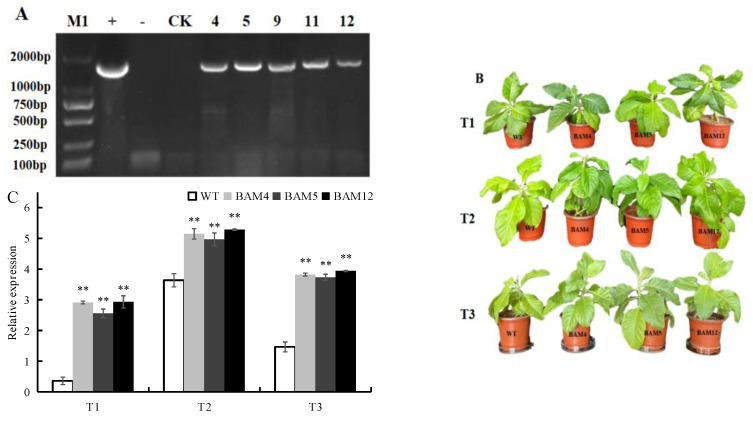
Cold tolerance testing of transgenic tobacco overexpressing *DoBAM1*. (**A**) PCR detection of *DoBAM1* gene tobacco. (**B**) Phenotypes of overexpressed *DoBAM1* tobacco and WT plants under control and cold treatments. (**C**) Transcript levels of *DoBAM1* under different conditions. M1 = DL2000 bp DNA marker; + = plasmid pPZP221–DoBAM1 as positive control; − = water as negative control; WT = wild type; 5–9 = different transgenic lines. ** indicate a significant difference at *p* < 0.01 compared with WT, respectively.

**Figure 8 genes-13-02296-f008:**
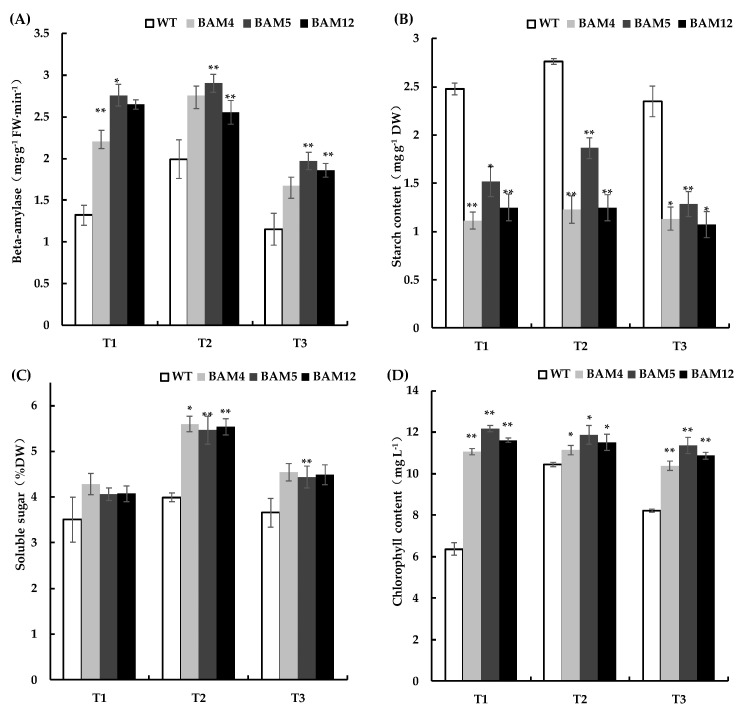
(**A**) β-amylase activity, (**B**) starch content, (**C**) soluble sugar, and (**D**) chlorophyll content in the DoBAM1-overexpressing lines and WT under different cold treatments. * and ** indicate a significant difference at *p* < 0.05 and *p* < 0.01 compared with WT, respectively.

**Figure 9 genes-13-02296-f009:**
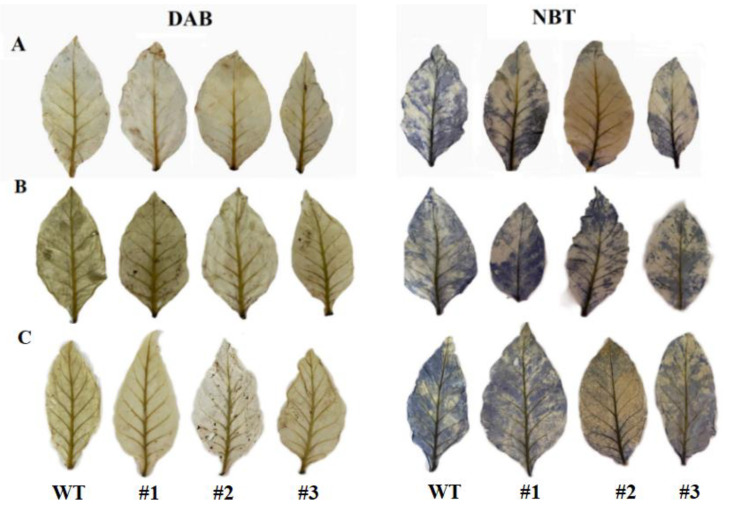
DAB and NBT staining of transgenic plant leaves is shown for plants subjected to cold treatments of (**A**) 4 °C for 2 h; (**B**) 2 °C for 2 h; (**C**) −2 °C for 2 h.

**Figure 10 genes-13-02296-f010:**
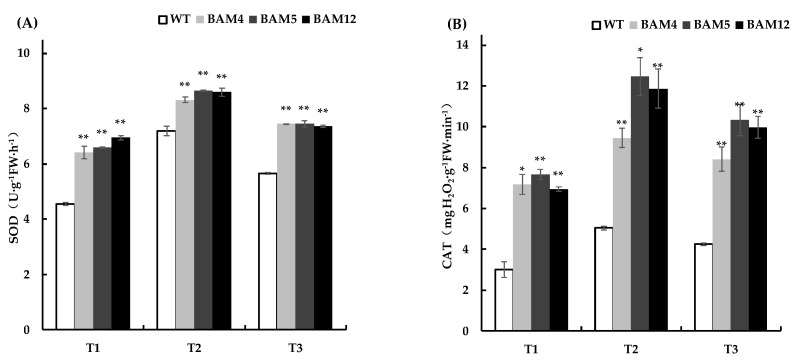
(**A**) SOD activity, (**B**) CAT activity, (**C**) POD activity, and (**D**) MDA content in the transgenic lines and WT plants under different conditions. * and ** indicate a significant difference at *p* < 0.05 and *p* < 0.01 compared with WT, respectively.

## Data Availability

Not applicable.

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
