# Peer review of "Overexpression of DoBAM1 from Yam (Dioscorea opposita Thunb.) Enhances Cold Tolerance in Transgenic Tobacco"

_genes, 2022, doi:10.3390/genes13122296_

Round 1

Reviewer 1 Report

The manuscript entitled “Overexpression of DoBAM1 from yam (Dioscorea opposita Thunb.) enhances cold Tolerance of Transgenic Tobacco” describe the isolation of a novel BAM gene called DoBAM1 from two yam varieties its expression pattern and overexpression in tobacco plants. The manuscript needs a major revision, especially for language and grammatical errors and after reconsideration, it can go for further evaluation. Moreover, at first glance title do not cover all the performed experiment in this manuscript as gene expression analysis has been also performed on yam plants. With other comments:

1- However, the manuscript is well-written but, in some parts, it needs a structural and grammatical language revision. Here are some examples:

L27: Yams (Dioscorea opposita Thunb.) belongYam (Dioscorea opposita Thunb.) belongs

L38: which is unique role → which has unique role or roles. However, this sentence (L38-L39) could be revised and rewritten. L39-41 is also a long and hard-to-understand sentence which need to be revised.

 L53: colding tolerance → cold tolerance

L110: The first part of the sentence lacks a verb L111: from data → from TAIR

L117: MEGA6 software ware MEGA6 software

L131: we were digested: You didn’t digest the vector was digested. Revise as: we digested

L143: was as described previously?!!

L143-144: wrong and incomplete sentence

2- The introduction is monotone and could be furnished with more information on the cold tolerance mechanism and the role of carbohydrate metabolism as there are some good research works in this field.

3- Please specify in material and method the exact origin of the studied material and how propagation materials have been obtained. Moreover, it has been notified that RNA has been extracted using a DNA extraction kit.

4- It has been addressed the sequence of used primer in Table S1, but there is no provided table!

5- Please also provide the cycle and temperature used for Real-Time analysis.

6- However B1 line has been used as a low starch cultivar but there is no difference between two analyzed cultivars based on starch quantification data.

7- In fig 3 no need to bring the size of all fragments of ladders. There are also some unspecific bands in the background. May it be due to two annealing temperatures (Please also provide the temperature used for PCRs)

 Altogether plenty of data-driven from various experiments have been provided in the present manuscript and it definitely deserves to be published in Gene providing addressing the risen questions.

Reviewer 2 Report

A very relevant paper that will be of interest to readers of the journal and not only in China.

Author Response

Thank you for giving me such valuable advice.

Reviewer 3 Report

I found this paper very interesting and with relevant application and ecological possibilities.

Some questions/comments to the authors:

- I think that tobacco was used just to facilitate the essays related to temperature. Is this correct? Why not using yam plants themselves? They would take too long to grow?

- In terms of the temperature essay, the authors think that it is realistic? Should it take longer, to simulate a possible adaptation by the plants? The increase in expression could be followed by a period of adaptation and decrease? Why the selected temperatures? Is there any ecological reason? It seems that this essay is somewhat artificial, in terms of the exposure conditions. It doesn't make ecological sense. Please explain. Why not increasing temperature to see a possible opposite reaction? Use temperatures closer to field conditions? The expression/activity only changes for these extreme conditions? I think a control with plants not exposed to cold conditions would be required to show that the increase in expression was due to cold exposure an not only to the different genetic make up of the trangenic plants, as compared to the WT.

- What would be the interest of this cold adaptation, ecologically, since the species of yam, I believe, mostly occur in relatively warm places? Moreover considering that global temperature is actually increasing.

- What would be the applied interest of producing cold adapted plants when the golbal temperature is actually increasing? Would they react similarly if the temperature decrease was smaller?

- Although the English seems to be quite reasonable, in some cases the text has some minor disconformities or errors. Please check again to ensure that the final version will be ready for a possible publication.

Round 2

Reviewer 3 Report

I kindly ask the authors to include their answers into the discussion of the paper and into other relevant sections.

Also include references where needed to better support those answers. 

This will be useful for the readers.

Regards,

LS

Author Response

I very much appreciate your comments. 

I have added the answers to your questions in the paper's introduction, Materials and Methods, and discussion, respectively, and cited the relevant references as support. 

I kindly ask you to review.